# A Short Tale of the Origin of Proteins and Ribosome Evolution

**DOI:** 10.3390/microorganisms10112115

**Published:** 2022-10-26

**Authors:** José Arcadio Farías-Rico, Carlos Michel Mourra-Díaz

**Affiliations:** Synthetic Biology Program, Center for Genome Sciences, National Autonomous University of Mexico, Cuernavaca 62210, Mexico

**Keywords:** protein, ribosome, evolution, peptide, oxygen, bacteria, folding

## Abstract

Proteins are the workhorses of the cell and have been key players throughout the evolution of all organisms, from the origin of life to the present era. How might life have originated from the prebiotic chemistry of early Earth? This is one of the most intriguing unsolved questions in biology. Currently, however, it is generally accepted that amino acids, the building blocks of proteins, were abiotically available on primitive Earth, which would have made the formation of early peptides in a similar fashion possible. Peptides are likely to have coevolved with ancestral forms of RNA. The ribosome is the most evident product of this coevolution process, a sophisticated nanomachine that performs the synthesis of proteins codified in genomes. In this general review, we explore the evolution of proteins from their peptide origins to their folding and regulation based on the example of superoxide dismutase (SOD1), a key enzyme in oxygen metabolism on modern Earth.

## 1. Introduction

There is no clear consensus on how life originated; however, there is general agreement that the building blocks of life were present under primitive Earth conditions [1]. Urey and Miller [2] have shown that early Earth conditions were sufficient for promoting the formation of primitive amino acids. It has been hypothesized that once amino acids—the building blocks of proteins—were present, they could polymerize into larger and more complex peptides simply through their interaction with clays [3]. The role of nucleotides at the beginning of life is also paramount, as such units are the basic functional elements that polymerized into RNA at the dawn of life [4].

Nature tinkered with RNA to overcome the problems of self-replication and storage of information [5]. Today, the ribosome stands as the most important molecular fossil that supports the existence of an RNA world. This nanomachine recognizes messenger RNA and translates its information into functional proteins. The conservation of ribosomal RNA is so prominent that it allows biologists to create phylogenetic trees of life [6] dividing the natural world into the Bacteria, Archaea, and Eukarya domains. Since its discovery by Nobel Prize winner George Palade in 1955, it has fascinated the scientific community to the point of pushing the discovery of its atomic structure. The 2009 Nobel Prize in Chemistry was jointly awarded to Venkatraman Ramakrishnan, Thomas A. Steitz, and Ada E. Yonath “for studies of the structure and function of the ribosome”.

Thus, structural biologists have been able to reconstruct the path from the ancient proto-ribosome [7] to the modern vertebrate-ribosome we observe today in humans [8]. The development of ribosomes allowed Darwinian evolution of a new metabolism in changing Earth conditions. In particular, the elongation and evolution of the ribosomal exit tunnel might have allowed the appearance of specialized enzymes to cope with environmental challenges.

One of the most prominent challenges faced by organisms on the early Earth was the so-called great oxidation event. The rise of photosynthetic cyanobacteria created a massive change in Earth’s atmosphere, namely resulting in molecular oxygen being raised to high levels; this was a massive threat for many organisms. In this context, proteins such as superoxide dismutase were an important adaptation. This enzyme transforms the superoxide radical into molecular oxygen and hydrogen peroxide. The superoxide radical is commonly produced as a byproduct of oxygen metabolism and can cause many types of cell damage.

In this review, we explore the current knowledge on how peptides might have arisen from chemicals. In addition, we look at how peptides could have associated with RNA in a process of chemical evolution to the point of assembling into functional ribosomes that allowed Darwinian evolution. We end with a discussion on how enzymes important for adapting to environmental changes, such as superoxide dismutase, are translated and regulated in bacteria. Our intention is that this review can provide general life scientists with sufficient information regarding how proteins could have arisen and be regulated in modern cells.

## 2. Early Chemistry on Earth and Protein Evolution

Geologists have confirmed that Earth is around 4.5 billion years old; shortly after its appearance (4.4 billion years ago), a protoplanet named Theia hit Earth, creating the Moon [9]. It was then that the Earth started to cool down and water could accumulate; records of the earliest fossil microbes found have been dated to around 3.8 billion years ago [10]. It is believed that life arose in a prebiotic soup of chemicals via a process called abiogenesis.

The prebiotic soup theory was published in a groundbreaking paper by Stanley Miller in 1953 [11]. Miller and his advisor Harold C. Urey designed an experimental setup to mimic the ocean and atmosphere of the primitive Earth [12]. In this experimental setup, they identified aspartic acid among other amino acids. Amino acids, however, are not the only molecules necessary for life; nucleotides are also required if Darwinian evolution is to take place by genetic mutation and adaptation. Juan Oro demonstrated that adenine (one of the building blocks of nucleic acids) could be created from hydrogen cyanide [13]. In summary, it has been demonstrated that two of the main building blocks for life could arise spontaneously from a prebiotic soup (Figure 1).

Along these lines, scientists have also proposed that amphiphilic molecules can be synthesized under plausibly prebiotic conditions and these molecules could assemble into protocells [14]. Jack Szostak has proposed that [15] lipid membranes could self-assemble into protocells and trap the nucleotides and peptides in such assemblies.

However, which came first: proteins or nucleic acids? Neither or both? It has been proposed that RNA could be the answer to the problem. RNA can perform similar functions to DNA, storing information but, in addition, also folding and catalyzing reactions [16] similarly to proteins.

The RNA world hypothesis, formulated by Carl Woese and Leslie Orgel in the 1960s, states that the most prominent molecule constituent of life was RNA or something chemically like RNA—i.e., a molecule that could replicate, transfer information, and additionally catalyze reactions. The existence of the ribosome (which we will review in the next section) is one of the most important pieces of evidence of this theory. The isolation of an RNA enzyme with RNA replicase activity provides support to the RNA world theory [17].

The role of peptides has seen a resurgence as they could be the central molecules of early life (Figure 1). This is based on several observations. Amino acids can be easily made in prebiotic conditions [18] and can react to create peptides [19]. Just recently demonstrated is the formation of thiodepsipeptide, a molecule that contains both peptide and thioester bonds. This is a plausible route for proto-peptide formation [20]. Proteins can also be made from noncanonical amino acids [21]. It seems plausible that proteins made from a reduced set of amino acids can reach folded states in conditions with high salt concentrations [22].

**Figure 1 microorganisms-10-02115-f001:**
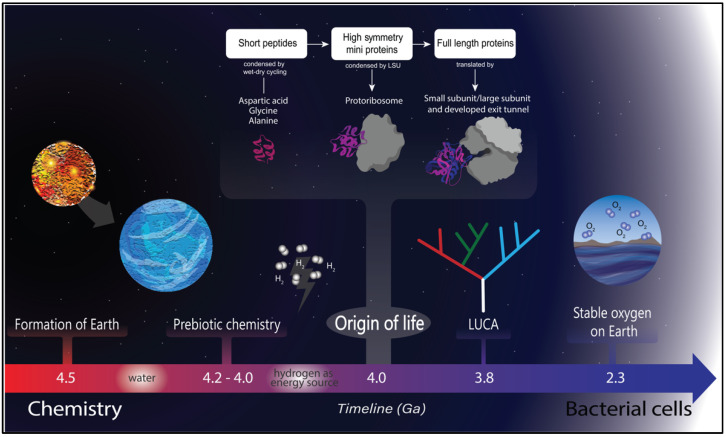
Timeline of life on Earth. Earth was formed around 4.5 Ga, and the presence of water is dated a short time later. Once water was available, a prebiotic chemistry was established, and hydrogen was a likely energy source [23] for life to originate. Life started around 4.0 Ga, characterized by coevolution between peptides (probably originated by wet and dry cycles) and RNA molecules, where the first proto–ribosome arose [24,25]. There are several lines of evidence for the last universal common ancestor at around 3.8 Ga [26], and a more recent event was the great oxidation of Earth that triggered a cascade of adaptations in the extant organisms [27].

The complication of the productive synthesis of long chains of nucleotides seems to pose a challenge to the normal understanding of the role played by peptides, making them more central in the prebiotic world. An interesting review proposed the coevolution of peptides and nucleosides in the early prebiotic world [24]. During the early periods of life, amino acids and their alternatives were either produced in the prebiotic soup (Miller experiments) or introduced to Earth by a meteorite. Later on, the first peptides could have arisen through condensation and wet–dry cycles [28]. These short and random peptides could start assembling in a process similar to the assembly of amyloid proteins, as is the case for SOD1 in Alzheimer’s disease [29]. In this case, the assembly of higher-order oligomers would be driven by thermodynamic control. In a world devoid of translation, it was likely that the first peptides aggregated into proteins in a similar process to amyloid formation, as happens today with proteins such as SOD1 in the context of ALS human disease [30]. This fact highlights the role of intrinsically disordered proteins in the origin of folded domains.

Intrinsically disordered proteins (IDPs) constitute between 25 and 30% of eukaryotic proteomes [31]. These kinds of polypeptides could represent an important step in the evolution of contemporary proteins [32]. In general, conventional modern proteins display low conformational heterogeneity in contrast to the numerous different states disordered proteins can display at diverse timescales. These proteins rapidly interconvert into different conformations taking various secondary structures, even at the level of coiled-coil structures.

Intrinsically disordered proteins do not fold because they display a biased amino acid composition. For instance, polar residues are overrepresented in their sequences, with hydrophilic residues and prolines being particularly overrepresented in IDPs, which, moreover, are usually depleted in aromatic hydrophobic residues [33]. These proteins could have been the most prominent polypeptides present at the dawn of life because of their simplicity and plasticity.

It is likely that, considering the evolution of the first proteins in abiotic conditions [34], the first functional peptides must have been short and not capable of complex folding. However, a peptide without function could not have been evolutionary selected. Therefore, RNA binding could have catalyzed the selection of short, disordered peptides. Short functional peptides composed only by residues synthetized under abiotic conditions could be the precursors of modern protein folds [34].

Along these lines, the “amyloid-first” theory states that if there were not disulfide bridges or aromatic residues, it is likely that the first small globular proteins or peptides would be present in a molten globular state and grow by aggregation. Intrinsically disordered peptides are one of the leading candidates for being the ancestors of folded architectures because they are simple and have repetitive amino acids that can be abiotically synthesized.

For instance, very few sequence changes would be needed to go from an unstructured intrinsically disordered peptide into a helix–hairpin–helix motif. This motif tends to dimerize in the presence of RNA, which supports the idea of an RNA world. This experiment demonstrated the transition from a flexible coacervate forming peptide to a stable structured protein domain with a specific function [35].

The appearance of translation allows the formation of more complex proteins from fragments. The next proteins could be small symmetrical domains that start to be translated by proto-ribosomes. At this level of protein evolution, where proteins also start assembling by repetitive secondary structures, the proteins would fold by thermodynamics in a funnel-like fashion, and this process would make them intrinsically refoldable [24,25,36].

The jump from chemical evolution to Darwinian evolution could have been explained by a coevolution of peptides and RNA that increased the reactivity and stability of ribonucleic complexes. The origin of the ribosome, for instance, which will be explored in detail in the next section, has been linked to the evolution of the genetic code and modern translation of complex proteins. For instance, the work of Ada Yonath, the 2009 Nobel Prize winner in Chemistry for her discovery of the structure of the ribosome, proposed a universal small RNA segment resembling a binding pocket called the “proto-ribosome”, which is present in all modern ribosomes. The authors synthetized such RNA pockets in the laboratory that were capable of mediating peptide bond formation [37].

The study of the evolution of folded proteins from peptides has been mainly conducted by examining determined protein structures. For instance, by comparing ancestral folds [38] researchers define sets of architectures that can be traced to the origin of the first enzymes: β-trefoil, TIM barrel, OB fold, or the immunoglobin-like fold are composed of recurrent super secondary structures, i.e., ββ hairpins, αα hairpins, or βαβ elements. Other authors argue that peptides must have formed nonspecific complexes that conferred them some adaptative advantage. Probably from here, in the RNA world, ribozymes were selected based on their ability to bind peptides. The available abiotic peptides were exhausted in the primitive soup and were not available any more because they were all binding RNA. A recent work has found an alphabet of 40 ancestral fragments where in the majority are related to RNA binding [39].

Novel studies on this matter have shown that these seemingly unrelated peptides are actually related in evolution [40], which makes it more probable that an ancestral set of peptides containing the 20 conventional amino acids could have existed in the Last Universal Common Ancestor (LUCA). Other researchers have also found that some precursors of proteins, the so-called P-loops in a single beta or alpha element, contain only early amino acids. The Dan Tawfik laboratory demonstrated that such motifs presented the activity of binding phospholigands and unwinding DNA [41]. The evolution of proteins from peptides is evident if we explore the most prominent superfolds in nature, proteins such as the TIM barrel fold. This fold is composed of repetitive βα_2_ units, and its evolution from fragments is comprehensively understood, therefore allowing the possibility to design a four-fold symmetric TIM barrel [42].

The next step in the evolution of proteins is the appearance of more complex domains that fold under kinetic control. Complex protein folds and multidomain proteins, such as the ones seen in eukaryotic proteomes, could have coevolved with the enlargement and development of the ribosomal exit tunnel [36] and accessory helpers such as chaperones [43]. We review in detail below the origin and evolution of ribosomal features.

## 3. The Origin and Evolution of the Ribosome

The ribosome is a ribonucleoprotein factory central in cellular metabolism [44]. These complexes perform protein synthesis in every cell in the biosphere; the ribosomes link amino acids together by reading the mRNA message produced by RNA polymerases. They are the decoding centers that allow the information contained in genomes to be translated into functional agents. The bacterial ribosome is composed of small and large ribosomal subunits: the large subunit is composed, in turn, of 5S and 23S RNA and several proteins, while the small subunit is composed only of the 16S RNA subunit. According to the RNA world hypothesis, this biomolecule is older than proteins. RNA has been demonstrated to be catalytic like current ribozymes [5]. At the beginning of life, it was necessary for a molecule to be autocatalytic and able to self-replicate.

The centrality of the ribosome was used by prominent biologist Carl Woese to propose a natural system of organisms, namely the three domains of life: Archaea, Bacteria, and Eukarya [45]. He proposed this system by comparing RNA sequences of the ribosomal rRNA 16S. Since their discovery by George E. Palade in 1955, ribosomes have been at the center of the work of many scientists. For instance, Venki Ramakrishnan shared the 2009 Nobel Prize in Chemistry with Ada Yonath for their work characterizing the ribosome structure. Ada Yonath’s important work continued by describing the origin of the ribosome at the peptidyl transferase center (PTC) [46]. Yonath and coworkers argue that the PTC of the modern ribosome is a universal symmetrical pocket solely made by RNA. It is furthermore argued that at the beginning of the history of the ribosome, the ability to bind peptides was useful for stabilizing the proto-ribosome.

Other scientists such as George Fox have investigated the connections inside the ribosome [47]. Fox argues that the ribosome was already present before the appearance of the last universal common ancestor or LUCA, and that the origin of the ribosome is located at the peptidyl transferase center (PTC), where the synthesis of proteins takes place. This corresponds well with the hypothesis of the RNA world that was mentioned in the previous section. Interestingly, Fox also mentions that early proto-ribosomes would be kept together with Mg^2+^. In another publication from the group of Hyman Hartman, it is also argued that the earliest region of the ribosome to appear would have been the large ribosomal subunit [48], with the decoding unit of the small ribosomal subunit being a later addition. The same research group also argue for a primordial piece of RNA that was able to fold and associate with small stabilizing peptides that formed the ancestor of the modern large ribosomal subunit.

Along the same lines, and with the purpose of describing the origin and evolution of the ribosome, Bokov and Steinberg developed a model of evolution of the 23S RNA [49].

They compared A-minor motif interactions in the IV domain of the 23S ribosomal RNA, basically removing elements and allowing the rest to remain intact, thus giving directionality in the growth of the RNA molecule. Later, Loren Williams and Anton Petrov developed the idea of insertion fingerprints, where branch and trunk structures are used to determine which parts of the ribosome are the most ancestral [50]. Basically, if one can define a helix and grow another one without perturbing the previous one, then the principle of dependency is developed.

Williams and coworkers argued that all ribosomes have a common core of bacterial ribosomes [50]. They described several phases of ribosomal evolution based on insertion fingerprints. As an example of the accretion model, they described the elongation of expansion segment 7. They described a stem loop of the rRNA from helix 25 and the descending rRNAs, generating a model of evolution of rRNA.

Helix 25 or expansion segment 7 originates in the LUCA, subsequently growing to 80 nucleotides in the ancestors of the eukaryote and archaea representatives. Interestingly, the Williams group managed to define where the insertion points are to propose relative ages of the RNA in the ribosome. The insertion fingerprints are defined as helical trunks linked to a secondary breaching helix at a specific junction [51]. The atomic fingerprints appear after comparison of pre- and post-insertion expansion sites. Examples of this are helices 52 and 38, which are common parts in *E. coli* that have branched in *S. cerevisiae*. Petrov et al. tested how insertion segments could be removed computationally, and the insertion removal could be healed by adding just one phosphate group. Subtle energy minimization can heal the wound.

Williams and coworkers dated the RNA that forms the PTC as the oldest biopolymer on Earth, while the ancestral expansion segment 1 (defined by the Williams method) is the ultimate ancestor of the PTC. This is joined by (ancestral element, AE) AES2 and AES3, but the sequence of joining events is not determined. The PTC is composed of AES1 to 5 and helices H74, H75, H89, H80, H90, H92, H73, and H93. Petrov et al. defined the following stages of ribosome evolution: Phase 1 = Folding and rudimentary binding activities, maybe catalytic activities. Phase 2 = Maturing of the PTC and exit pore. Phase 3 = Early tunnel extension. Phase 4 = Acquisition of the SSU interface. Phase 5 = Acquisition of translocation function. Phase 6 = Late tunnel extension. Phase 7 = Eukaryotic segments are acquired in simple eukaryotes. Phase 8 = Surface elaboration in complex eukaryotes. In a following publication, Kovacs et al. expanded their work to the evolution of ribosomal proteins [52] and managed to explain how proteins were added to the ribonucleoprotein ribosomal complex. Because they managed to assign directionality in RNA evolution, they then managed to date proteins as well. Many ribosomal proteins extend from a globular domain to the inside of the ribosome with longer loops (Figure 2).

The protein regions are defined according to contact made in different phases. For instance, in Figure 2, the protein part colored in green is in contact with RNA from phase 3. The part in yellow contacts RNA from phase 4, and the last part of the protein, which is a more globular and modern, is colored in red. Alva et al. argue for a coevolution of protein folding together with the ability of RNA to fold.

Other authors have also viewed the ribosome as a fossil record [54]. These authors also discussed the structures of the ribosomal proteins as being less structured the deeper the proteins go into the center of the ribosome. Moreover, the parts of the domains that are more exposed to solvent tend to present a more globular structure. There is an RNA function to chaperone proteins to fold. The Lupas laboratory made an important discovery by creating a soluble protein from the disordered helical ribosomal domain RPS20 [55].

Interestingly, there is no universal consensus on the temporality and directionality in the evolution of ribosomal parts. As has been discussed, many scientists agree on the ribosome originating in the PTC, but there is one group that suggests a different story. Using a cladistic phylogenetic approach, Caetano-Anoles and coworkers argue that the most ancestral part of the ribosome lies in the processivity center. According to their methods, the ribosome originated in the structures that support RNA decoding and ribosomal mechanics (the ratchet, Figure 3). While an almost universal consensus defines the PTC as the most ancestral part of the ribosome, Caetano and coworkers argue that the most ancestral part is the processivity center of the ribosome (mRNA decoding and mRNA helicase).

Caetano-Anoles applied sophisticated methods to hierarchically organize RNA molecules based on cladistic principles [56]. He derived universal phylogenetic trees based on rRNA secondary structures. The main conclusion from his analysis is that components of the small subunit that oversee ribosomal processivity evolved earlier than the peptidyl transferase center [57].

In a subsequent publication, the Williams group defined the evolution of the ribosome at atomic resolution [50]. As previously discussed, the Williams group developed a higher-resolution version of a model proposed five years before. Bokov and Steinberg studied the A-minor motif of the ribosome, an abundant tertiary structure interaction in the large subunit [49], to develop a model in which shells of RNA domain could be removed to deconstruct the structure.

Conceptually, for Steinberg and Bokov, the ribosome performs two functions: the selection of the RNA and the transpeptidase reaction (we believe that the exit tunnel performs a third yet unrecognized folding-associated function). RNA forms the core of the ribosome, and proteins are only around it to stabilize it. The core of the ribosomal RNA is very similar among all organisms; therefore, it must have formed before the split of the three kingdoms. Steinberg and Bokov studied the tridimensional structure of the 23S ribosomal RNA. In this region, the RNA folds with a particular pattern called the A-minor motif. Later, by removing shells from the ribosome (conceptualized as an onion in Figure 4), Steinberg and Bokov identified 19 elements (or expansion segments) that could be removed without affecting the integrity of the remaining part. They argued that the addition of shells could have happened easily with every shell locked only if it made the previous structure more stable and kept the activity of transpeptidase.

The resolution of the above discussed model was improved by Williams and Petrov. They managed to define the mechanism by which the expansion segments grow; for instance, inserting an elongation helix into a more basal helix. They identified molecular indications of these sites as insertion fingerprints (using atomic-resolution ribosome structures). Armed with the mechanism of how the expansion segments grow, they extrapolated backwards to define the most ancestral RNA segments and locate the origin of the ribosome at the PTC.

In the next chapter of the debate [59], Caetano-Anoles criticized Williams’ approach, saying it was inductive in nature with no mathematical or phylogenetic method to support it (such as maximum parsimony or Bayesian reconstruction). Caetano-Anoles also argued that assessments of trunk branch assumptions are incorrect. He indicated that correct trunk-branch directionality of properly assigned ancestral insertions of expansion segments suggest an origin of the large rRNA molecule in ribosomal mechanics and not in the PTC. Caetano closed this chapter by saying that Williams’ models are nomothetic in nature (a tendency to generalize) and not ideographical; in other words, they do not analyze differences in numerous organisms.

Later in the discussion, the Williams group [60] named Caetano’s approximation as GIGO (growth inferred by genothermal ordering), and the criticisms from Caetano were addressed one by one. The main argument by the Williams group was that there is a consensus on the PTC as the most ancestral part of the ribosome [46,47,49,50,61,62,63,64,65]. They also challenged Caetano’s argument that an increase in RNA stability was a primary driver of evolution. The characteristics of RNA (stability, length of helices, etc.) were used as phylogenetic characteristics by Caetano to create trees.

Linear RNA segments are more ancestral than branched rRNA according to the GIGO model. However, the Williams group say that Caetano’s assumption is falsified, using as evidence long linear GC-rich helices in eukaryotic ribosomes. These insertions could be random novel acquisitions that in turn would be more stable because of the GC content. The partition of the RNA is also criticized as being subjective.

The consensus analysis, according to Caetano, is based on the study of few structures, and it falsely supports the origin of the ribosome around the peptidyl transferase activity. Caetano differentiates his method as historical, while he defines the consensus as nomothetic. It is possible, however, that the approaches used by the two different groups in dispute are both ideographic and nomothetic. It is generalized that structures and sequences can be compared and superposed, and these elements are studied with mathematical approaches that are ideographic or historical. In the most recent publication of the debate [66], Caetano proposes a way out of the controversy, where he writes, “historical and nonhistorical scientific methods” are integrated.

All the discussions presented by Williams and Caetano also point toward the importance of the exit tunnel. By adding a concentric piece of RNA on the top of the ribosome from the center, the graduality of the ribosomal exit tunnel growth is evident. It is likely that a longer tunnel would allow the synthesis of more complex proteins. Two more relevant publications shed some light on the controversy of what is the most ancestral part of the ribosome. In the first, the ribosome is conceptualized as the first self-replicating biological entity before the first cells [67]. In this theoretical paper, the authors present evidence of a self-replicating machine in the ribosome. The sequences of the ribosomal RNAs are not only structural but also codify for proteins related to proto-ribosomal metabolism. They codify for replication enzymes, such as polymerases, and it is stated that tRNAs could have arisen from the rRNAs via fragmentation. RNA could thus be a vestige of an ancient genome that codified for a self-replicating and autocatalytic intermediary with the modern catalytic activities we observe today. The same authors, in a follow-up publication, argue for a “ribosome-first” theory of evolution [68].

Another interesting publication that profits from the newly available structures of the ribosomes demonstrates that the classical tree of life can be reconstructed from the pure geometry (Figure 5) of the exit tunnel [58]. The authors derive a metric that describes the geometric features of the ribosomal exit tunnel of many species among the three kingdoms of life. With this approach, they managed to define radius variation plots that are clustered to differentiate and properly allocate the branches of the three domains of life.

Changes in the exit tunnel might indicate changes in the metabolic activity of cells. It is likely that a longer tunnel would have allowed the synthesis of more complex proteins to cope with cataclysmic challenges during evolution on Earth, e.g., events such as the great oxidation event.

## 4. Great Oxidation Event, Expansion of the Amino Acid Repertoire, and the Superoxide Dismutase Enzyme

Approximately 2.4 billion years ago, the rise of molecular oxygen posed a great challenge for life on Earth. This event is known as the great oxidation event (GOE) [69,70]. It is possible to obtain approximate dates for when this event took place based on organic biomarkers and isotopic evidence, such as through red beds, lateritic paleosols, and records of the mass-independent fractionation of sedimentary sulfur isotopes [71].

During the Paleoproterozoic period around 3.2 Ga ago, our planet had an atmosphere rich in nitrogen, carbon dioxide, and methane, with euxinic oceans characterized by high availability of iron and sulfur (Figure 6a) [72]. This early environment would have enabled anaerobic organisms to proliferate freely and to develop metabolic mechanisms to take advantage of the elements that were present in high concentrations. Nevertheless, this situation changed with the arrival of cyanobacteria 2.8 Ga ago. These bacteria evolved the ability to consume sunlight and convert carbon dioxide and water into sugar, releasing oxygen as a waste product in a process called photosynthesis. The success of this mechanism is heavily influenced by the availability of suitable electron donors that are vulnerable to oxidation.

This process represented a great advantage for cyanobacteria, since it gave them the ability to produce their own energy. The rest of the species of early Earth took a considerable time to adapt to the new oxygenic conditions [73]. This is mainly because the result of the water oxidation process also entails the production of reactive oxygen species, which can be found in different forms (O_2_, H_2_O_2_, and OH). These forms are highly toxic for the molecular machinery of organisms.

**Figure 6 microorganisms-10-02115-f006:**
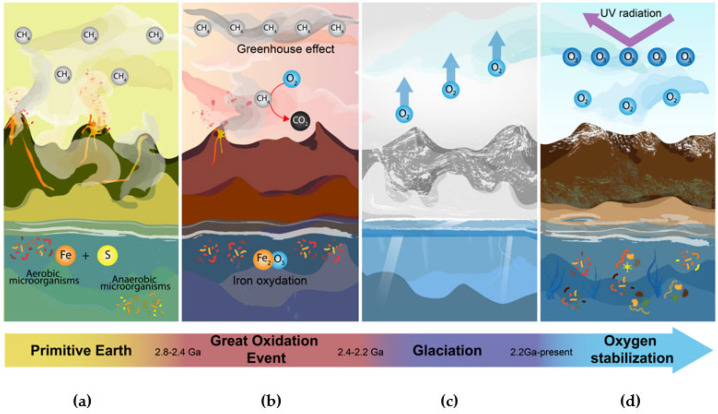
Transitions of the great oxidation event. (**a**) Primitive Earth. On early Earth, oxygen concentrations were very low. The atmosphere contained a large proportion of methane, while the oceans and their deep trenches had high availability of elements such as iron and sulfur [74,75,76]. (**b**) Great Oxidation Event. Cyanobacteria began to produce oxygen as a byproduct of photosynthesis. Oxygen bioprocessing created reactive species that were highly toxic to organisms and, in turn, began to interact with oceanic iron and atmospheric methane. This caused the generation of iron oxide, turning the seas red, and a gradual increase of oxygen and carbon dioxide in the atmosphere [72,74,77]. (**c**) Glaciation. The removal of methane from the atmosphere caused the existing greenhouse effect to be lost. This caused a dramatic drop in temperature freezing the entire planet, killing a wide variety of species [72,77]. (**d**) Oxygen Stabilization. The continuous production of oxygen caused ozone to accumulate in the atmosphere and form a layer that retains UV radiation. This favored the emergence of organisms capable of taking advantage of the available oxygen, which regulated the levels of reactive species in the atmosphere [75,78].

While the cyanobacteria were capturing solar energy to feed themselves, they were simultaneously poisoning the rest of the organisms that shared their environment [69]. Furthermore, the oxygen release promoted the formation of harmful chemical products. When cyanobacteria started releasing oxygen, this triggered several chemical reactions with iron in the oceans, which led to the formation of iron oxide and, later, the change in the color of the oceans to red (Figure 6b). This event caused water to become toxic for many species.

With its continuous release, oxygen started accumulating in the atmosphere, resulting in the oxidation of methane into carbon dioxide. The gradual disappearance of methane in the atmosphere led to a sudden cooling that finally produced a glaciation event that is currently known as the “Paleoproterozoic snowball Earth” (Figure 6c) [71]. However, although the great oxidation event was catastrophic for many organisms, it also led to the accumulation of ozone in the atmosphere, which formed a protective layer against deadly ultraviolet rays. As a result, this created the perfect environment for the proliferation of life as we know it today.

The rise of organisms that adapted to the new Earth conditions was made possible by the development of mechanisms allowing them to resist and take advantage of atmospheric oxygen, causing the levels of this gas to eventually stabilize (Figure 6d).

If we consider all the adaptations that organisms on early Earth required to survive the GOE, we could explain how aerobic metabolism of bacteria arose, and how modern organisms found a way to give this toxic component a central role as a key element in processes, such as signal transduction [69]. Reactive oxygen species (ROS) can cause cell damage in three different ways.

It has been observed that ROS are capable of damaging cell membranes by the well-known process of lipid peroxidation (Figure 7A). Reactive oxygen species can also attack genetic material, resulting in DNA strand breaks, which are harmful for the organism because the repair of these breaks can lead to deleterious mutations (Figure 7B). Finally, ROS have the potential to oxidize proteins, so this could result in a malfunction of many cell processes, such as disruption to receptors causing a signaling pathway to stop functioning (Figure 7C) [79].

Eukaryotes adapt the use of ROS as a defense mechanism against pathogenic bacteria; for instance, phagocytes produce ROS by the activation of NADPH oxidase (Figure 7d) [80]. However, in evolution and the constant race of adaptations, bacteria have implemented mechanisms that protect them against the eukaryotic antimicrobial system. These responses consist of the activation of transcriptional factors such as OxyR, PerR, and SoxRS, which are responsible for activating the antioxidant defense system. In this response, the activation of key enzymes such as catalase, thioredoxins, glutathione reductases, ferric uptake regulator, and superoxide dismutases is necessary [80].

As alternatives to these adaptations between pathogens and hosts, there are several pathways that use ROS as key elements in eukaryote signaling. For instance, nitric oxide plays a central role in the regulation of many functions such as vascular tone, nerve function, and immune regulation (Figure 7c). Another example is demonstrated by diatomic oxygen, because this compound plays an important role in the intracellular signaling of tumor necrosis factor-α (Figure 7d) [79].

It was not possible for organisms to evolve mechanisms to defend against ROS without changes in protein structure and function. For example, several studies show an evolutionary leap thanks to the implementation of ROS detoxifying enzymes such as NADPH oxidases, peroxidases, and superoxide dismutases (SODs) [81]. SOD proteins are responsible for the dismutation of diatomic oxygen to hydrogen peroxide, and are of great importance for living beings, which is reflected in their high degree of conservation since the times of early Earth [82]. In humans, we can see the importance of the protein SOD1 in the case of amyotrophic lateral sclerosis (ALS).

This disease attacks upper and lower motor neurons, provoking fatal muscle paralysis. The studies on this disease have been key in the gradual discovery of the functionality and conformation of SOD1 proteins.

A dysfunction in the SOD1 protein has been found to play a pathogenic role in ALS by inducing harmful effects in a misfolded form [83].

To date, evidence suggests that conservation of SOD1 proteins (Figure 8) could be because this type of protein is among the original enzymes found on early Earth. One piece of evidence supporting this is that one family of SOD enzymes possesses an ideal catalytic center for the union of Fe with its prosthetic group. The SodB enzyme can use iron, which is consistent with the conditions of the euxinic oceans on early Earth. The use of this metal ion would be consistent with the high concentration of Fe and S available. Currently, there are three families of SOD, which are defined according to the metals used for their catalysis. Each of the families comprises some isoforms of the enzyme and contains the following heavy metals: Fe or Mn, Cu and Zn, or Ni [82]. In the next section, we will explore the folding and regulation of SodC.

## 5. SOD1 Co- and Post-Translational Folding and Regulation

Molecular oxygen dramatically changed the conditions in which proteins operate. Oxygen plays a monumental role in protein folding, as demonstrated in the milestone Anfinsen experiments [86]. These experiments were conducted via oxidative refolding of the unfolded RNAse I. The initial process of protein unfolding, which involves the reduction of native disulfide bridges, could allow refolding to the native state via formation of the correct four bridges under oxidative conditions [87].

The Anfinsen experiments are a classical example of in vitro studies. These experiments were performed with purified protein in diluted buffer conditions. The only other molecules present during the folding reaction, besides the protein, were salts and the buffer system. The folding of proteins in the cell presents a completely different scenario; for instance, there is interplay between the ribosome and the folding of its products during biosynthesis (Figure 9).

In recent years, the ribosomal exit tunnel has taken the spotlight for its importance as a novel active site that is under evolutionary pressure to properly fold proteins. Many cellular processes occur in the ribosomal exit tunnel while proteins are still being translated: formation of protein complexes [88], chaperone binding [89], enzymatic processing [90], and disulfide bond formation [91].

Disulfide bond formation in protein folding is a complicated process, especially for proteins that are heavily influenced in their function by formation of the correct disulfides. As an example of such a process, we will briefly explore the folding of superoxide dismutase SOD1, a key enzyme in oxygen metabolism, as it has been described in the previous section. CuZn superoxide dismutase (SOD1) is an important protein in combating oxidative stress. SodC from *E. coli* displays binding sites for copper and zinc; interestingly, these sites are not universally conserved in all SOD1 proteins [92].

Conversely, SOD1 displays an intramolecular disulfide bond in all species, which suggests that this bond is essential for its folding and function. The maturation process is not clearly understood in bacteria.

SOD1 is important for all organisms; in eukaryotes, its deletion causes a reduction in human lifespan, and mutations in *H. sapiens* are linked with amyotrophic lateral sclerosis [93]. In pathogenic bacteria, SOD1 is involved in protection against a respiratory burst of the host immune system [81], as outlined in the previous section.

It has been shown that SodC (*E. coli*, SOD1) folding is directed by formation of the disulfide bond [92]. This protein does not show much secondary structure in the absence of the disulfide bond, in contrast with the eukaryotic SOD, where the beta barrel folding pattern is maintained even if the disulfide bond is present in a reduced state; but its dimerization state is affected [94]. Some studies in bacteria have shown that the formation of the disulfide bond is necessary for binding of metal ions at the zinc binding site, while the human SOD can bind metals without the disulfide bridge. However, the steps of maturation are not yet clear for the bacterial counterpart. The synthesis process and maturation of bacterial SOD1 is complex: the protein must be translocated into the periplasm by the SecA system. It displays a signal peptide that expands for the first 19 residues. The disulfide bond is formed in the protein, once in the periplasm, leading to binding of its metals and, finally, the characteristic beta barrel is formed.

There is little information on protein folding thermodynamics or kinetics for SodC. However, there is the following information about maturation [95]: the apo form of SOD is the most unmatured version of the protein; the unfolded protein is probably cotranslationally translocated into the periplasm via “standard” secretory signal peptides transported by the Sec translocon and cleaved by signal peptidase I (Lep). Once in the periplasm, where there is a higher concentration of zinc and copper ions, the protein binds either copper (Cu^2+^) or zinc (Zn^2+^) via the canonical binding site for copper (and the cysteines that form the disulfide bridge in the mature form). In a subsequent step, the protein forms the disulfide bridge while the copper is still bound. In a final step, the protein binds zinc in the second binding site [95].

The disulfide formation in SodC is a complex topic, and there is evidence that in knockouts of the Dsb (disulfide bond) formation system, the disulfide bond still forms in a self-catalytic manner. There is one crystal structure of CuZnSOD from a Bacteroidetes bacterium that shows hybrid characteristics (prokaryotic and eukaryotic features). From this information, Wright proposed a mechanism for self-catalyzed disulfide maturation [96]. This special protein seems to provide evidence that supports the endosymbiotic origin of eukaryotes.

Lynch et al. showed how the apo SOD, together with the zinc-loaded form, unfolds faster than the copper loaded and the fully loaded form with both ions. Lynch et al. demonstrated as well that the binding of copper is more important than the binding of zinc alone for kinetic stabilization of the protein. As expected, the full holo version is the slowest unfolder [97].

**Figure 10 microorganisms-10-02115-f010:**
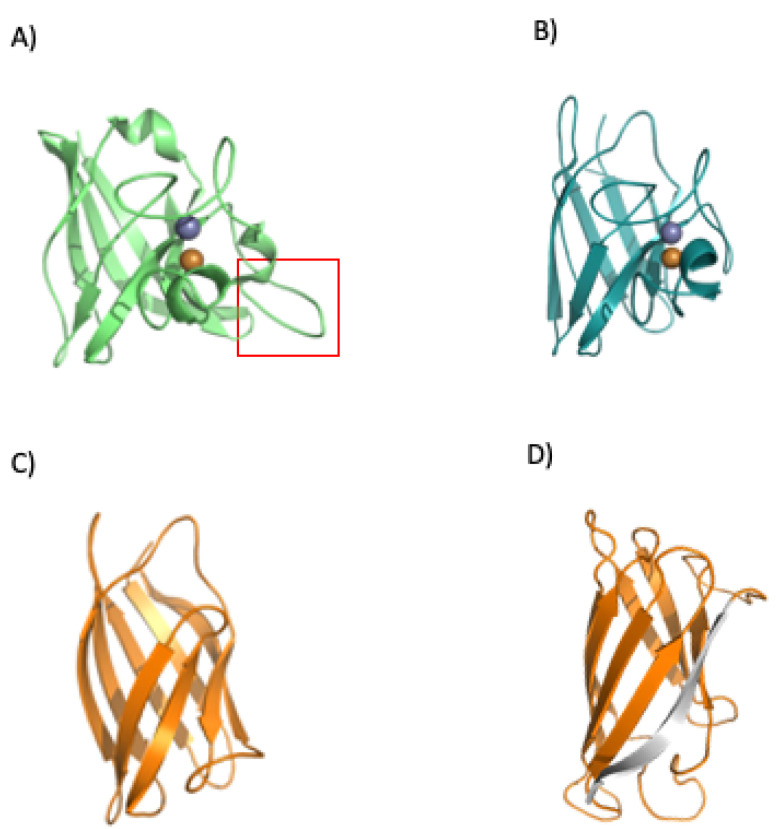
Folding and regulation of SOD1. The bacterial SOD displays a loop insertion in the metal binding area (**A**) red rectangle, which makes the protein more prone to misfolding when there is no binding of its cognate metals. The human protein does not show the loop (**B**) (PDBid 1spd), and a loopless version of the human protein (without the metal binding area, panel (**C**) was analyzed in our previous work (**C**) [98]. The analysis showed a probable folding intermediate that lacks the terminal beta (shown in gray, panel (**D**)).

The group of Mikael Oliveberg performed detailed work with superoxide dismutase 1 from *H. sapiens* [94,99]. They found that native eukaryotic SOD is a symmetric homodimer, whereby each monomer coordinates one catalytic copper and one structural zinc (the copper still also helps to increase the kinetic stability of the protein). The zinc binds to loops IV and VII, which are critical for dimerization. This process is controlled by metal binding and disulfide bridge formation.

To understand how the loops affect the structural features of the protein, they were removed, and the cuts rejoined using short linkers. This removal (in the eukaryotic human SOD) disrupted the dimer interface and created a protein that is soluble and monomeric. The engineered protein displays a robust structure (Figure 10C). Loop removal increased the folding rate and decreased the unfolding rate constants. In addition, an overall increase in thermodynamic stability was observed [99].

Many of the folding experiments performed on SOD1 have been conducted in vitro and, therefore, we decided to study how ribosomes could influence protein folding. We took the noLoops (4bcz) protein version and created a force profile using arrest peptides [98] in a cell-free system. We aimed to discover how the barrel would form during translation.

Force sensors are based on short peptide sequences that arrest translation while being synthesized by the ribosome [100]. The chemical energy generated by the protein during folding is converted into mechanical energy that releases the arrest. The method provides a proxy for the structuring process of the protein during translation. By using a force sensor together with a limited proteolysis assay, we demonstrated that SOD1 (no loops version, PDBid: 4bcz) displays a folding peak when the protein is located 45 residues away from the peptidyl transferase center. The force profile shows an increase in the tension of the chain around 25 residues away from the PTC, and at this distance from the center of the ribosome, the terminal beta strand of the protein is not available to establish interactions. If this is the case, the small increase in folding force would indicate a folding intermediate. This folding intermediate would comprise the full barrel without the terminal beta, and the limited proteolysis affects this intermediate because it could only be marginally stable, and we only see full protection from proteolysis at around L-40 (40 residues away from the ribosomal center). To gain more insights into the folding mechanism of SOD1, we performed bioinformatic predictions on three different SOD1 versions from two kingdoms of life. We used the %MinMax software developed by the laboratory from Patricia L. Clark (University of Notre Dame) [101] for the detection of rare codon clusters and performed free energy cotranslational folding calculations using a Python bioinformatic pipeline [102].

Our protein models for the analysis were: (a) SOD1 from *Escherichia coli* (PDBid 1eso), (b) the full version of human superoxide dismutase (PDBid: 1spd), and (c) monomeric human CuZn superoxide dismutase (PDBid: 4bcz). The analysis with the noLoops version of superoxide dismutase displayed a cluster of rare codons at the end of the ORF together with a drop in free energy (Figure 11 and Figure 12). These results are indicative of a folding intermediate that could lack the terminal beta strand. This is in accordance with our previous experimental force measurements on the ribosome. The same analysis was performed with a full version of human SOD (PDBid: 1spd), showing no differences with the noLoops version. The main difference is that the full protein is longer, because of the presence of the loops. In general, however, it displays the same rare codon conservation and the same likely intermediate at the end of the protein.

We observed a different scenario when we performed the analysis with SOD1 from *E. coli*. In this case, the start of the protein displayed rare codons. The program seems to have predicted a cluster at the beginning of the protein, but it is more likely related to the signal peptide present at the N-terminal part of the protein.

In general, these analyses predict a folding intermediate that is consistent in Bacteria and Eukarya. However, the measurements are not sensitive enough to detect the differences in folding caused by loop removal. The example of SOD demonstrates general trends in protein evolution, where stable scaffolds are embellished with loops that can hold metals to regulate dimerization and functional features.

## 6. Conclusions

In this review, we have explored the possible path from chemicals to proteins. Primitive Earth conditions would have allowed the emergence of random peptides that, in turn, assembled into larger biomolecules. The world of RNA could have been assisted since the start by peptides; at some point, these two essential biomolecules formed a proto-ribosome that exhibited two features essential for life: storage of information and self-replication. Once there was a ribosome, there could be translation and Darwinian evolution, the landmark processes by which organisms adapted to environmental challenges such as the great oxidation event. Along these lines, enzymes such as superoxide dismutase 1 play important roles in helping organisms to cope with the current high levels of oxygen.

## Figures and Tables

**Figure 2 microorganisms-10-02115-f002:**
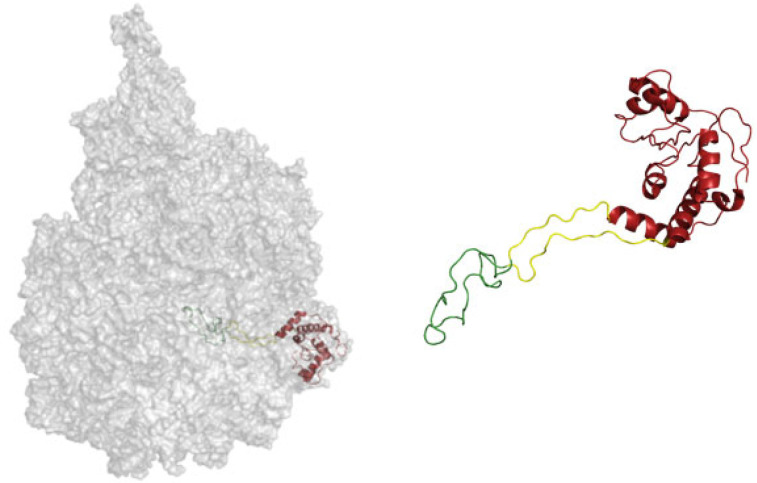
Different evolutionary phases mapped on the structure of the ribosomal protein L4. The ribosomal protein contacts RNA from different epochs, and the most internal coil (green) makes contact with the most ancient RNA. Then, the yellow part is structured around newer RNA, while the red region is the most derived [53]. A similar characterization is proposed by Alva et al., where proteins are taken as a whole and the cytoplasmatic complement is related to the most outer shell of the ribosome [54].

**Figure 3 microorganisms-10-02115-f003:**
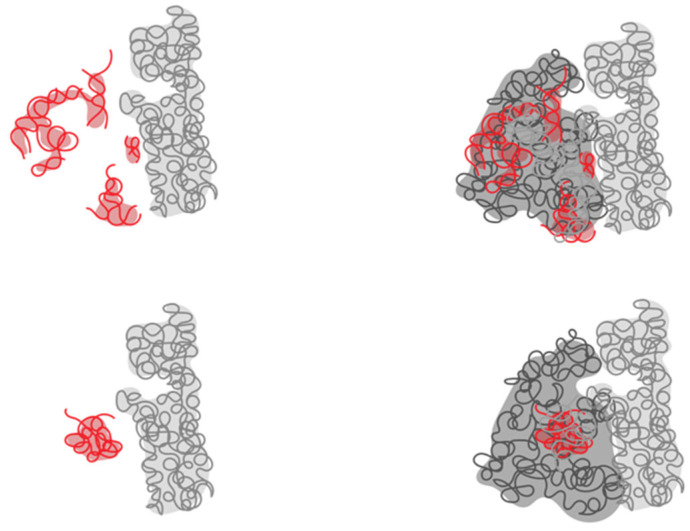
Abstraction of the ribosome showing discrepancies in defining the most ancestral part of the ribosome. According to the consensus (bottom), the most ancestral part is the PTC (in red), while according to Caetano-Anoles, the most ancestral part is in the processivity region (in red, top).

**Figure 4 microorganisms-10-02115-f004:**
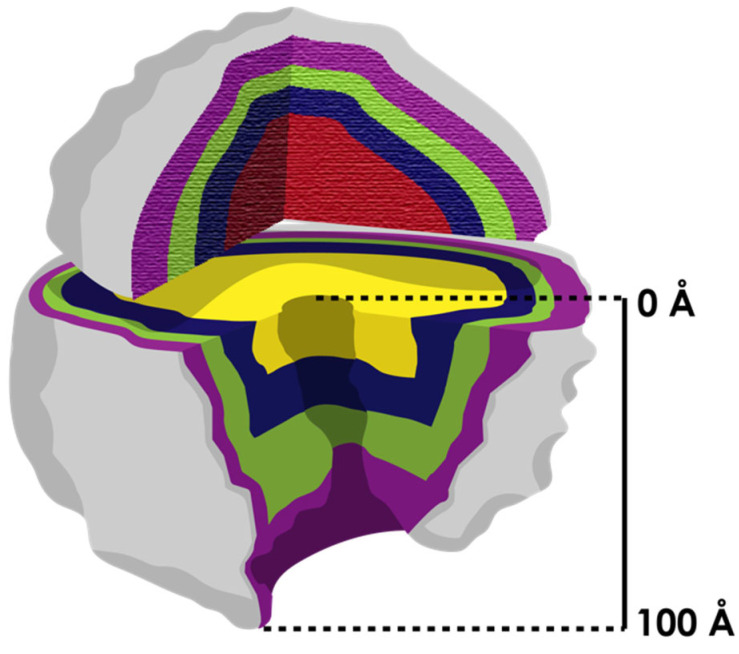
The ribosome is conceptualized as an onion (with layers indicated in different colors) that grows from the center (the most ancestral part). The two subunits evolved independently (most inner shell in different color) at first, and the process of coevolution occurred later when they associated. According to Williams [36], the only structure that evolved continuously on the ribosome was the exit tunnel (it extends over 100 Å from the PTC to the vestibule). Other authors [58] have demonstrated that differences in the exit tunnel increase among organisms from the center to the surface.

**Figure 5 microorganisms-10-02115-f005:**
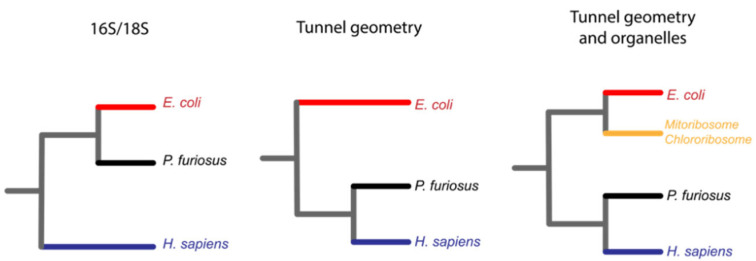
The geometry of the exit tunnel recapitulates the classical phylogeny of life made by sequence comparisons of 16S ribosomal RNA [58]. The classical phylogenetic tree of life is made by comparison of ribosomal 16S/18S RNA, and it allowed the creation of the three domains of life, Bacteria in red, Archaea in black and Eukarya in blue (left). Duc et al. codified into a single metric the geometry of the ribosomal exit tunnel and clustered it to recreate the three domains of life in a similar fashion (center). The same authors also included the geometry of the organelle ribosomal exit tunnels that grouped in the bacteria branch of life (right).

**Figure 7 microorganisms-10-02115-f007:**
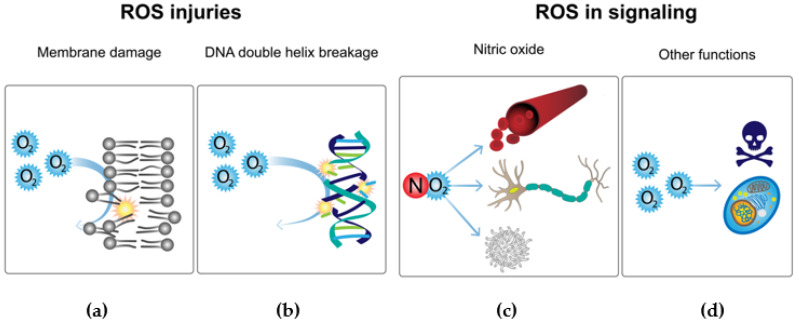
Toxic ROS effects and implementation in signaling pathways. Comparison of the damaging effects of reactive oxygen species and their current implementation in oxygen metabolism. (**a**) Reactive oxygen species (ROS) can have highly toxic effects on living beings, one of the most harmful being the damage to cellular membranes by lipid peroxidation [79]. (**b**) ROS also can attack DNA, resulting in strand breaks [79]. (**c**) Nitric oxide plays an important role in the regulation of several cellular functions [79]. (**d**) ROS contribute to the intracellular signaling of tumor necrosis factor-α [79].

**Figure 8 microorganisms-10-02115-f008:**
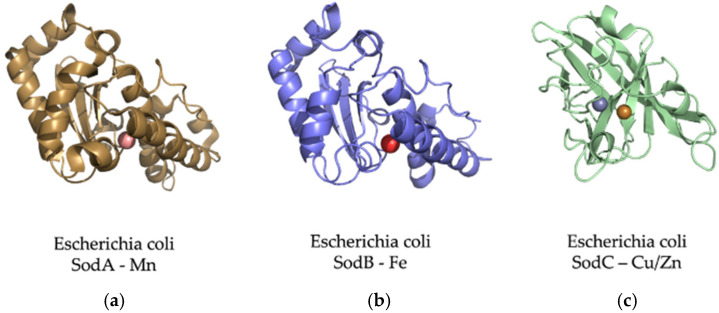
The adaptation of SOD proteins to different metal ions depending on their abundance. The SOD enzymes can convert the oxygen free radical (O_2_^−^) into hydrogen peroxide (H_2_O_2_) and water (H_2_O) [84]. (**a**) *E. coli* Mn-superoxide dismutase protein. MnSOD is very important to protect cells from the dangerous effects of overproduction of ROS [85]. It is a homotetrameric enzyme that has manganese at its active center to achieve an efficient O_2_ dismutation. (**b**) E. coli Fe-superoxide dismutase protein. Several works consider FeSOD as the oldest of this type of protein. This is because its emergence in primitive organisms is consistent with the high bioavailability of iron. It shares 42% of its identity with the MnSOD amino sequence [82]. (**c**) CuZnSOD proteins are known as the most modern family of the SOD lineages. These enzymes are present in Bacteria and Eukarya but not the Archaean domain, and for this reason, it is thought that this family is more recent than those of iron and manganese [82].

**Figure 9 microorganisms-10-02115-f009:**
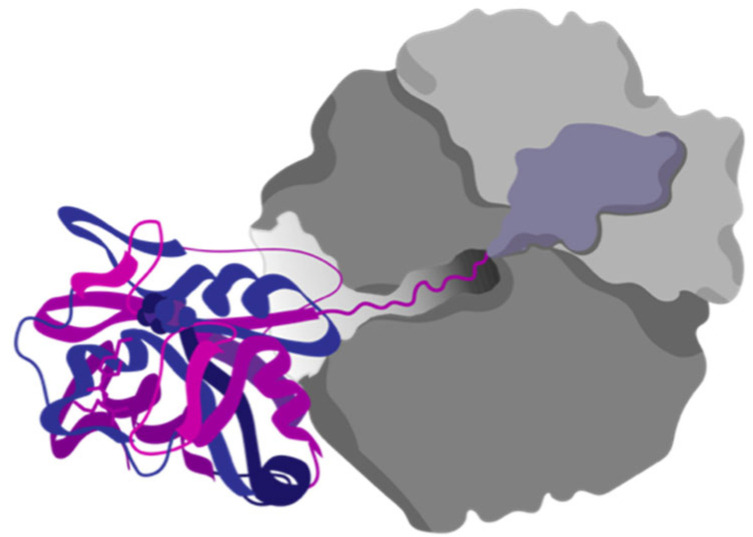
Protein cotranslational folding. Proteins can start to fold while being translated by the ribosome. The discovery of cotranslational folding has strong evolutionary implications for the ribosomal exit tunnel.

**Figure 11 microorganisms-10-02115-f011:**
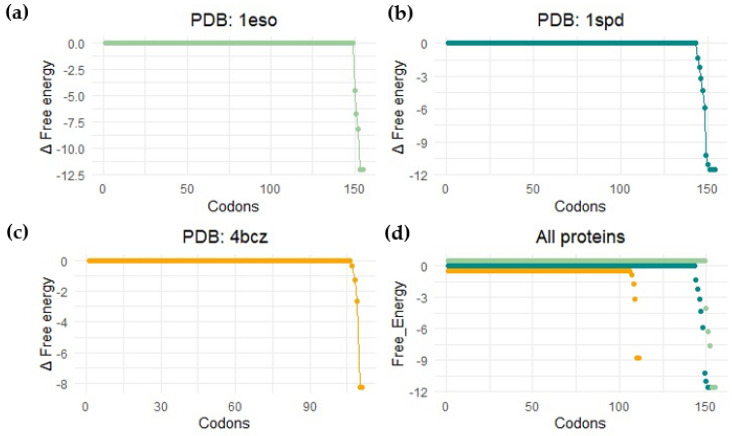
Bioinformatic detection of folding energy release. This image shows the free energy calculation results by performing a coarse-grained calculation of cotranslational folding (software published in [102]). Profile for (**a**) *Escherichia coli* SodC (Superoxide dismutase Cu-Zn, PDB: 1eso); (**b**) human-related amyotrophic lateral sclerosis CuZn superoxide dismutase (PDB: 1spd); (**c**) monomeric human CuZn superoxide dismutase in which loops IV and VII are deleted (PDB: 4bcz); (**d**) all proteins.

**Figure 12 microorganisms-10-02115-f012:**
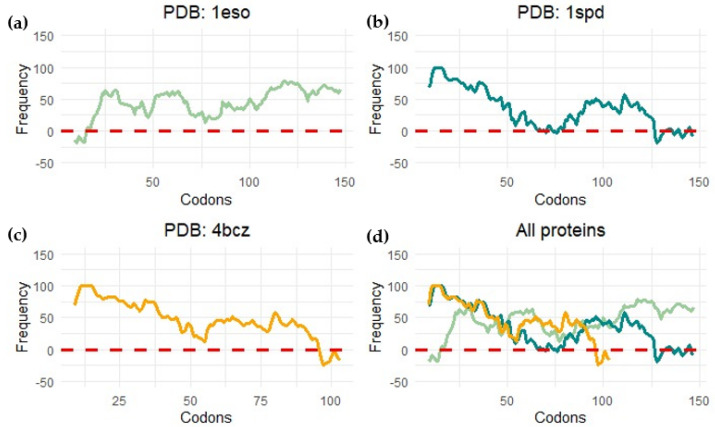
Rare codon cluster detection. This image shows the measure of codon frequency in three different proteins by using the %MinMax calculator (software published in [101]) for (**a**) *Escherichia coli* CuZn superoxide dismutase (PDB: 1eso); (**b**) human amyotrophic lateral sclerosis CuZn superoxide dismutase (PDB: 1spd); (**c**) monomeric human CuZn superoxide dismutase in which loops IV and VII are deleted (PDB: 4bcz); (**d**) all proteins.

## Data Availability

Not applicable.

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
