# Peer review of "A Short Tale of the Origin of Proteins and Ribosome Evolution"

_microorganisms, 2022, doi:10.3390/microorganisms10112115_

Round 1

Reviewer 1 Report

This is an important and informative review that will be of interest to many readers and will have a noticeable impact. Although I appreciate the efforts of authors to consider the role of the evolution of proteins in evolution of life, their focus on foldable proteins represents the major limitation of this work. This is because they completely ignore the phenomenon of protein intrinsic disorder (i.e., the absence of unique structure in functional proteins and regions). In fact, it is recognized now that many proteins in different organisms do not have unique structures. Even more proteins contain functional disordered regions. It is also clear that disordered proteins were at the roots of the protein evolution, as abiotic synthesis of a structure-less polypeptide is more probable than the appearance of a protein with unique structure. Furthermore, disordered proteins/regions are typically characterized by poor sequence conservation. Therefore, they cannot be easily analyzed using existing phylogenetic tools. Consideration of these points should be included to the manuscript.  

Minor issues:

1) It is accepted practice to use "amino acid" instead of "aminoacid"

2) On line 276 it is stated: "discovery by George E. Palade in 195,..." 

3) For the first time, term PTC was used in line 267, but it was defined later, on line 290. 

4) The manuscript contains some grammatic errors and careful editing and proofreading is required. 

Author Response

Dear reviewer one,

We thank for your comments. We have included a whole new discussion regarding intrinsically disordered proteins as you suggested. We definitively missed this important point in our previous version of the manuscript. Moreover, we have subjected our manuscript to a thorough English revision by the English editing service of the MDPI journals, we hope we had corrected all the minor issues mentioned (we attached the certificate provided by the team in case you need to see it). 

Reviewer 2 Report

In this paper, the authors tried to explain how simple molecules evolved to functional proteins.  However, they failed to show the origin of amino acids, peptides, and RNAs.  Particularly, their description on the formation of amino acids and peptides looks somehow what was published in 1970s. 

I agree that SOD is a quite important enzyme for survival of all the terrestrial organisms.  It was believed that SOD was essential before GOE.  They tried to show that less than 20 amino acids were necessary after GOE, but it is widely accepted that LUCA used 20 amino acids.  As a whole, this review would not "provide to the general life scientist with a source of information."

Thus, I do not think that this paper can be published in the present construction.

Author Response

Dear reviewer 2,

Thanks for your comments, we have made significant changes in the manuscript. We hope that our new MS construction feels more modern and can be informative in the new version. We have corrected the imprecision regarding the 20 aminoacids and the LUCA. Moreover, we have subjected our manuscript to a thorough English revision by the English editing service of the MDPI journals. We attach the English certificate in case you need to see it.

Reviewer 3 Report

The review article entitled " A short tale of proteins: from chemicals and peptides to folding and regulation" is an effort to present the information regarding the "history" of proteins for life scientists. In my point of view the scope of the present review paper to help life scientists understanding the evolution of proteins could not lead to a scientifically interesting contribution. Even though the literature used is appropriate and quite new the present review paper does not fit to the scope of the journal. 

For the above mentioned reasons the present manuscript does not merit publication in Microorganisms. 

Author Response

Dear reviewer 3, we thank you for your comments,

We have made significant changes to the structure of the manuscript and have subjected it to a thorough English revision by the English editing service of the MDPI journals. Regarding your comment of the manuscript to be unfit for the scope of the journal we could argue that we discussed oxygen metabolism in bacteria. We also took the opportunity to discuss ribosome evolution and evolution of proteins from peptides. Even if some topics could lie a bit outside of the scope of the issue, we consider that our MS could help life scientist to initially explore important topics in protein evolution. We attach the certificate of the English editing service in case you need to see it.

Round 2

Reviewer 1 Report

All critiques were adequately addressed and the manuscript was revised accordingly. I do not have new concerns.

Author Response

We thank the reviewer comments and we are glad there are no more concerns

Reviewer 3 Report

The manuscript has been substantially improved and can be accepted in its improved version if the Editorial Team finds it suitable for this Special Issue. 

Author Response

We are thank the reviewer comments and we are glad our MS can be considered for publication